# A Review on Liquid Crystal Polymers in Free-Standing Reversible Shape Memory Materials

**DOI:** 10.3390/molecules25051241

**Published:** 2020-03-10

**Authors:** Zhibin Wen, Keke Yang, Jean-Marie Raquez

**Affiliations:** 1Laboratory of Polymeric and Composite Materials Center of Innovation and Research in Materials and Polymers, Materia Nova Research Center & University of Mons, 23 Place du Parc, B-7000 Mons, Belgium; 2Center for Degradable and Flame-Retardant Polymeric Materials (ERCEPM-MOE), National Engineering Laboratory of Eco-Friendly Polymeric Materials (Sichuan), State Key Laboratory of Polymer Materials Engineering, Sichuan University, Chengdu 610064, China

**Keywords:** liquid crystal polymers, shape memory materials, reversible strain, thermal responsive, photo responsive

## Abstract

Liquid crystal polymers have attracted massive attention as stimuli-responsive shape memory materials due to their unique reversible large-scale and high-speed actuations. These materials can be utilized to fabricate artificial muscles, sensors, and actuators driven by thermal order–disorder phase transition or *trans*–*cis* photoisomerization. This review collects most commonly used liquid crystal monomers and techniques to macroscopically order and align liquid crystal materials (monodomain), highlighting the unique materials on the thermal and photo responsive reversible shape memory effects. Challenges and potential future applications are also discussed.

## 1. Introduction

Liquid crystal polymers (LCPs) are of intense interest due to their unique anisotropic shape changing and mechanical properties [1]. During the past decade, researchers are examining and discovering their fascinating properties, such as rapid and large reversible actuation in order to make this class of materials good candidates for stimuli-responsive reversible shape memory materials [2,3,4]. Compared to traditional shape memory materials, one of the most remarkable properties of LCPs is a fully reversible, equilibrium phenomenon. It also has astonishingly large amplitude (200–300%), and can be stimulated by temperature change or irradiation by light [5]. The possibility of applications ranges from actuators and sensors [6,7] to artificial muscles [8] and active smart surface [9,10,11]. In this review, order–disorder phase transition of the thermal or *trans–cis* photoisomerization mechanisms of shape change of LCPs will be introduced in Section 2. We summarize several pathways to prepare monodomain LCPs in Section 3, highlighting the unique materials applied to free-standing reversible shape memory effects in Section 4.

## 2. Mechanisms of Shape Change in LCPs

The basic principle behind the shape change of LCPs depends on alignment directions and relevant phase transitions (nematic, cholesteric, smectic, or isotropic) [3]. The majority of studies on the actuation of LCPs based on thermally induced order–disorder phase transition are illustrated in Figure 1a [2,12]. Materials exhibit anisotropic deformation along the director orientation as the order parameter (S) change above the phase transition temperature. For another photoinduced type, such as azobenzene-containing LCPs, the stabilized rod-like *trans* azobenzene mesogens transfer to unstable bent *cis* isomers irradiation with UV light, resulting in the decrease of the order parameter. Then, the volume shrinkage in the film surface cause bending behavior (Figure 1b) [13]. Based on the mechanisms, the widely studied monomers are adopted to fabricated LCPs illustrated in Figure 1c [2,3,11,14,15].

## 3. Strategy for Preparation of Monodomain LCPs

To observe free-standing reversible actuation, macroscopic alignment of the mesogens is necessary to form a LC monodomain. Up to now, a variety of approaches has been used to generate uniformly macroscopic LCPs. This section will discuss the commonly used methods.

A two-step crosslinking technique is one of the most commonly implemented and simplest methods to dictate alignment. Multifunctional groups are designed in monomers to fabricate a network controlled by react ratio [16,17,18]. In the initial step, a weakly crosslinked elastic network is formed and aligned by mechanical stretching of the polymer chains, against the entropy. The liquid crystal phase is further locked in the conformation of the backbone by a following second crosslinking step. Küpfer and Finkelmann introduced a fast and a slow crosslinking process for polysiloxane-based LCPs. An optically clear film was fabricated and fixed with an external load during the second crosslinking stage. The highly ordered monodomain of the sample has been proven by the X-ray experiment [16]. However, it should be noted that a high internal stress in the network and a limited ordering of the mesogens inhibited this approach, because the first stage required a partly crosslinked network to facilitate subsequent programming.

External fields alignments are the useful methods to prepare LCPs. Typically, the low viscous liquid crystal monomers are melted and cooled into liquid crystal phase, then aligned by external fields (such as surface rubbing [19,20], photo alignment [21,22], electric [23], or magnetic alignment [24,25]. Finally, the mesogens are fixed by further polymerization. The direction of liquid crystal relative to a substrate surface plane is critically determined by the nature of the surface. Rubbing on polyimide alignment layers has been used to fabricate a monodomain LCP film [26,27]. The monomeric mixtures are melted on a glass substrate coated with a rubbed polymer film (polyimide, poly (vinyl alcohol)), cooled into liquid crystal phase parallel to the rubbing direction, and polymerized. This kind of photoalignment was first reported by Ichimura et al. in 1988 [28]. A substrate surface is modified with photochromic units (such as azobenzene) in molecular levels to alter the chemical structures and molecular orientation by irradiation. The alignment of the mesogens is controlled by photochromic reactions such as *trans–cis* photoisomerization [29,30]. Ferroelectric liquid crystal monomers possess a very strong dipole moment. The materials prefer to be parallel to the direction of electric field, orienting the monodomain. An anisotropic film can be obtained by crosslinking under an appropriate field [31,32]. The monodomain LCP film can be synthesized by the use of magnetic fields for the alignment of nematic liquid crystals [24,33]. Due to strong diamagnetism, the mesogens are aligned along the orientation of applied magnetic field when cooling to the nematic phase from isotropic phase. Based on this basis, many methods were recently developed for monodomain LCPs including the electrospinning process [34], microfluidics [35,36], inkjet 3D printing [37,38,39], and soft lithography [40,41]. Despite much success in implementation of the aforementioned external assisted methods, monodomain LCPs also present unique challenges such as limited sample size, the inability to reprogram, or complex 3D structure.

More recently, to break through these bottlenecks, researchers put forward a new strategy by synthesizing monodomain LCEs with dynamic covalent bond exchange, called covalent adaptable networks (CANs), or vitrimers [42,43,44]. Vitrimers are able to change their topological structure due to the exchange reactions under certain external stimuli. To date, most of the monodomain LCPs with exchangeable links were prepared based on transesterification [45,46,47], transcarbamoyalation [48], boronic-ester bond [49], disulfide [50,51], and allyl sulfide groups [52,53,54]. Summarized in Figure 2, Ji et al. demonstrated that after programming the LCEs monodomain with the topological network rearrangement were facilitated by thermo-induced transesterification. The material showed excellent reversible shape memory effects and 3D shape change [47]. Bowman et al. reported a spatiotemporal control of alignment based on photoactivated allyl sulfide bond exchange, which represents a powerful way to create thermal reversible freestanding films [52]. Although precious structures and external stimuli in the arrangement are required in the process, CANs serve as a powerful tool to achieve monodomain LCPs.

## 4. Shape Change of LCPs Triggered by Different Stimuli 

As already described in Section 2, thermal and photo are primary stimuli used to actuate shape change of monodomain LCPs. The phase transition driven by direct heating is the simplest and easily available stimuli, which has been widely observed in LCPs. In many cases, light emerges as a contactless energy source as it can be directed for microscale devices from rapidly turned on or off and spatially remote distances [55], including photo-heat effect and *trans–cis* photoisomerization. In this review, the ratio of shape change is calculated to the original stain of samples.

### 4.1. Thermal Responsive LCPs

Thermal responsive LCPs exhibit shape change along the director orientation as the order parameter changes by the heating above and cooling below the phase transition temperature. Contraction and expansion behaviors of LCPs driving by direct heating and cooling along the aligned orientation director is shown in Figure 3. The first reported monodomain LCP was prepared by Finkelmann et al. using a two-step crosslinking method depending on the react speed of C=C double bonds of vinyl groups and methacrylate groups to the Si-H of polyhydrosiloxane chain. A shape change in length of 90% parallel to the alignment direction was obtained during the nematic–isotropic phase transition [16]. A reversible actuation of an LCP film can lift up and put down a 10 g weight when heating and cooling (Figure 3a) [56]. In our previous work, we fabricated a monodomain LCP synthesized by a commercial liquid crystal monomer RM82 via thiol-Michael reaction. This material shows independent isotropic−nematic and nematic−smectic phase transitions. A single system possessed two-way reversible strains and multiple shape memory effects (Figure 3b) [48]. In Figure 3c, Ratna et al. reported side chain polyacrylate LCPs that exhibit strains of 35–45% to the stresses of 210 kPa. These muscle-like materials have potential possibility to design artificial muscle actuators (compared to muscle shrinkage stress of 300 kPa) [27]. In addition to the macro-sized system, micro- or nano-sized actuators are desirable for practical application. Keller et al. utilized a soft lithography technique to prepare micron-sized responsive liquid crystal pillars [35,57]. The pillars underwent a reversible contraction and expansion in the order of 30–40% when heating and cooling (Figure 3d). A similar micrometer-size actuator was shown in Figure 3e synthesized by Zental et al. via the microfluidic method. The particles consisting of monodomains showed a reversible change in length of about 70% [36].

Aligned LCPs exhibit dimensional changes along the alignment direction in response to temperature changes. However, the implementation of programmable shape change in applications requires the further development of soft materials that exhibit spatial, predictable, and complex reversible variations. Some precious works have been reported by researchers, such as patterned approach and 3D printing [58,59]. In Figure 4a, White et al. prepared a spatially heterogeneous liquid crystal elastomers with surface alignment and optical patterning methods. They demonstrated a variety of three-dimensional programmable actuation and shape changes [60]. Photoinduced dynamic exchange processes enable spatial resolution and patterned reprogramming in any topography state, such as allyl dithiol. Bowman et al. prepared a diacrylate oligomers by thiol-Michael addition reaction and then created complex, predictable, and spatially reversible shape changes by programming LCP in the LC phase or isotropic phase (Figure 4b) [53,61]. Nowadays, the 3D printing technique offers new scope to construct complex topography. Lewis et al. printed a various of complex 3D LCPs that resulted in reversible shape change (Figure 4c) [39] and the transformed shapes can be locked on demand with photo-activated dynamic bond [54].

In addition to the direct heating to actuate the shape change of LCPs, light is employed to trigger the shape changes in a precise and remote control. Light is a contactless energy source as it can be directed for microscale devices from rapidly turned on or off and spatially remote distances [55]. A noncontact actuation by doping the LCP matrix with a light-heat transfer agent to trigger the order–disorder transition of mesogens is summarized in Figure 5. These photothermal materials include graphene [62,63], carbon nanotubes [64,65], conjugated polymers [66], chromophore [67], organic dyes [68], and gold nanocrystals [45,69]. 

In Figure 5a, Terentjev et al. prepared a series of LCPs mixed with carbon nanotubes (CNTs). They demonstrated that CNTs can absorb infra-red/visible light and convert it into local heat, thus triggering thermal response of LCPs [70,71]. Yang et al. synthesized a monodomain polyaniline nanoparticle/LCP composite with respect to the NIR-stimulated photo-activated performances. The material exhibited a fully reversible shape memory effect when turning on or off the NIR light [66]. Taking advantage of the extraordinary photothermal conversion property of NIR chromophore, an ultrafast photo responsive speed soft actuator has been reported by Yang’s group. The material can raise the local temperature from 18 to 260 °C in 8 s, and lift up its 5600 times weight irradiation by 808 nm NIR light (Figure 5b) [67].

### 4.2. Photo Responsive LCPs

Apart from thermal responsive LCPs, photo-driven LCPs containing azobenzene mesogens are typical kinds of materials that have been widely used for fabricating shape change actuators. The reversible *trans–cis* photoisomerization leads to a tiny decrease in the order parameter of LCPs which contribute to the shrinkage in the film surface. Figure 6 shows a variety of behaviors which has been explored such as bending and unbending [26,72,73,74], twisting [75], oscillations [76,77,78], and three-dimensional movements [14,79]. 

Ikeda and Yu et al. synthesized the first azo-LCP film exhibited bending behavior in response to the chosen direction by linearly polarized light (Figure 6a) [72]. The new 3D movement of azo-LCN composite materials was also reported by the same group. Based on this concept, a new photomechanical nanoscale plastic motor was developed to convert light to mechanical energy directly by simultaneous irradiation with UV and visible light at room temperature (Figure 6b) [14]. White et al. prepared azobenzene-containing monolithic LCPs (azo-LCNs) exhibiting photoinduced motion irradiation with UV–visible light. The twisted-nematic orientation transforms the films from flat to spiral ribbons, resulting in complex controllable photomechanical behaviors (Figure 6c) [75]. With the concept of the reversible change of order parameters of azobenzene units, Broer et al. designed a smart chiral nematic surface coating exhibited reversible change in the surface topology upon exposure to UV and visible light (Figure 6d) [11,80], A sponge-like coating has been developed that exhibited the ability to release and absorb a liquid after irradiation with light. The surface forces between the coating and an opposing surface can be controlled by light, resulting in tunable adhesion [81]. More recently, an interesting light-driven robot was prepared by Xia and Zhao et al. This AuNRs hybrid azobenzene LCP combined two photo responsive mechanisms of the *trans–cis* photoisomerization of azobenzene exposure to UV and LC–isotropic phase transition by AuNR’s NIR photo-heat effect, showing preciously controllable motions (Figure 6e) [45]. 

Considering the safety, cost, and power consumption, high-energy light UV is not an ideal source for practical applications in the real world. Researchers are enthusiastic to design visible/NIR light responsive systems. Yu et al. designed a novel molecule excited triplet−triplet annihilation based on upconversion luminescence (Figure 7a). They doped the low-power activated mesogens into azo-containing LCP matrix to create a photo responsive composite. The film achieved red and NIR-light triggered bending via an emission-reabsorption process [82]. Furthermore, they constructed robust tubular LCP micro-actuators, which exhibited asymmetric topologic variation irradiation with intensity gradient visible light, resulting in capillary forces for liquid propulsion (Figure 7b) [83]. Fluorinated azobenzene molecules have been reported that can be switched solely by visible light [84]. To realize the application in sunlight, Broer et al. described that the LC soft actuator doped with a visible light responsive ortho-fluoroazobenzene moiety exhibited continuous self-propelling oscillatory motion upon exposure to sunlight shown in Figure 7c [77,78].

## 5. Conclusions

Compared with traditional stimuli-responsive materials, LCPs are capable of fast and large-scale reversible shape change, which enable the distinctive potential applications in artificial muscles, actuators, sensors, and robotics. The key to generate spatial actuation in LCP materials is the fabrication of monodomain. The alignment of mesogens toward monodomain has to be achieved through several methods. Dynamic covalent bonds provide a promising approach to program the orientation of mesogens and break through the limitation of permanent shapes. Combined with the recently developed techniques will provide a promising opportunity to design programmable LCP actuators. The preciously and controllable diverse range of spatial variation in directionality will further broaden these stimuli-responsive materials research. One interesting aspect of LCPs is soft robotics that responded to the surrounding stimuli exhibiting autonomous motions, such as oscillating, rotating, rolling, or twisting. Another is the recently smart surface showing morphological changes to enable tunable surface wettability, reflective color, adhesion, and haptic actuation. To develop a “real-life” application of LCPs, the development of patterned methods of LCPs and the low threshold actuation of stimuli-responsive materials will require a multidisciplinary approach in theories and technologies.

## Figures and Tables

**Figure 1 molecules-25-01241-f001:**
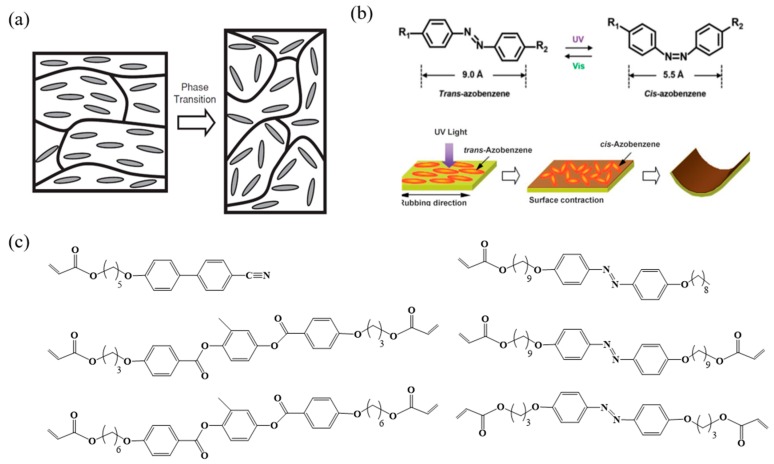
(**a**) Mechanisms of liquid crystal polymers (LCPs) shape change of the polymer chain suffering from anisotropic deformation to its coiled conformation above the nematic-to-isotropic phase transition. Reproduced with permission from [2]. Copyright © 2010 WILEY-VCH. (**b**) Mechanisms of the photoinduced bending behavior in the azobenzene-containing LCPs caused by a *trans–cis* photochemical phase transition in the surface. Reproduced with permission from [13]. Copyright © 2006 WILEY-VCH. (**c**) Chemical structures of common liquid crystal monomers [2,3,11,14,15].

**Figure 2 molecules-25-01241-f002:**
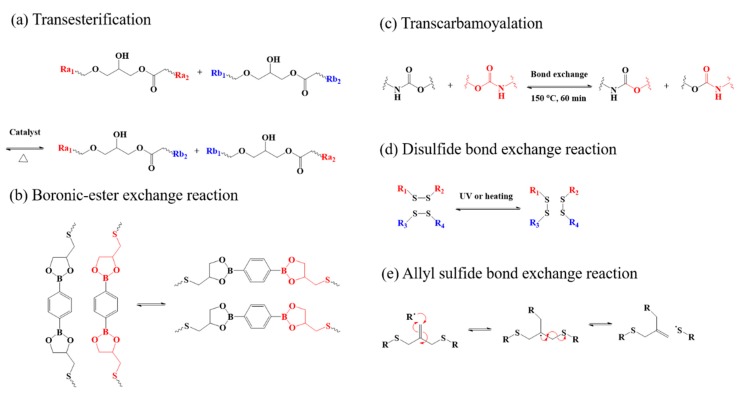
Dynamic covalent exchangeable bonds in LCPs. (**a**) Transesterification. [47] (**b**)Boronic-ester exchange reaction [49]. (**c**) Transcarbamoyalation [48]. (**d**) Disulfide bond exchange reaction [50]. (**e**) Allyl sulfide bond exchange reaction [53].

**Figure 3 molecules-25-01241-f003:**
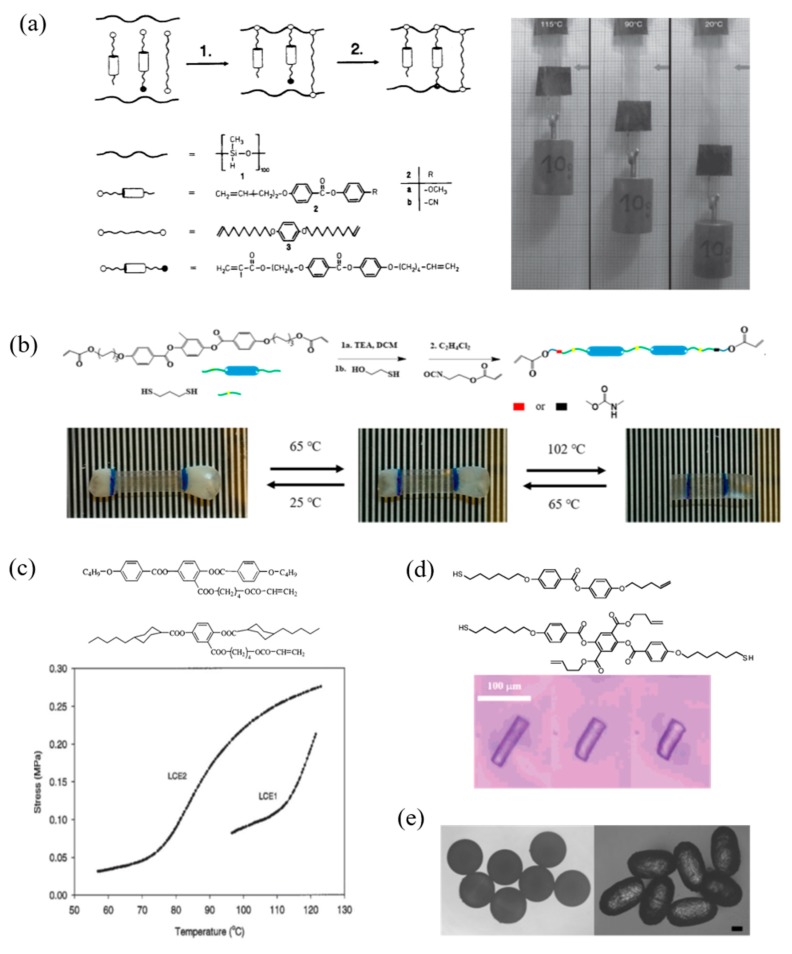
Contraction and expansion behaviors of LCPs along aligned orientation director. (**a**) Schematic illustration of a two-step crosslinking method and a reversible strain can lift up and put down a 10 g weight. Reproduced with permission from [16]. Copyright © 1991 WILEY-VCH. (**b**) The two-stage reversible strains were achieved by smectic A−nematic and nematic−isotropic phase transitions upon thermal cycling (from left to right). Reprinted with permission from [48]. Copyright (2018) American Chemical Society. (**c**) Iso-strain measurement on side-chain LCPs heating through the nematic to isotropic phase transition. Reprinted with permission from [27]. Copyright (2001) American Chemical Society. (**d**) An isolated pillar exhibits a contraction with the order of 35%. Reprinted with permission from [35]. Copyright (2006) American Chemical Society. (**e**) LC-particles possess reversible changes from spherical to cigar-like. Reproduced with permission from [36]. Copyright © 2009 WILEY-VCH.

**Figure 4 molecules-25-01241-f004:**
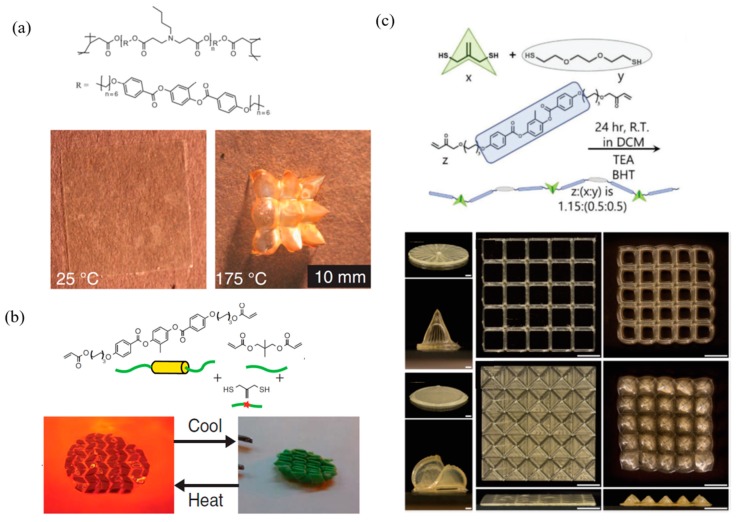
Predictable and complex reversible variations of LCPs. (**a**) LCP films prepared with a 3 × 3 array of +1 radial defects can be actuated to generate periodic topographical surfaces. Reproduced with permission from [60]. Copyright © 2015, AAAS. (**b**) The polymer was folded, fixed, and light-induced reprogrammed with 320 to 500 nm resulting in unfolding at high temperatures and folding during cooling. Reproduced with permission [53]. Copyright 2018, The Authors, some rights reserved; exclusive licensee American Association for the Advancement of Science. Distributed under a CC BY-NC 4.0 license, published by American Association for the Advancement of Science. (**c**) Programmable shape morphing LCPs fabricated by the 3D printing possess capable of reversible actuation. Reproduced with permission [39]. Copyright © 2019 WILEY-VCH.

**Figure 5 molecules-25-01241-f005:**
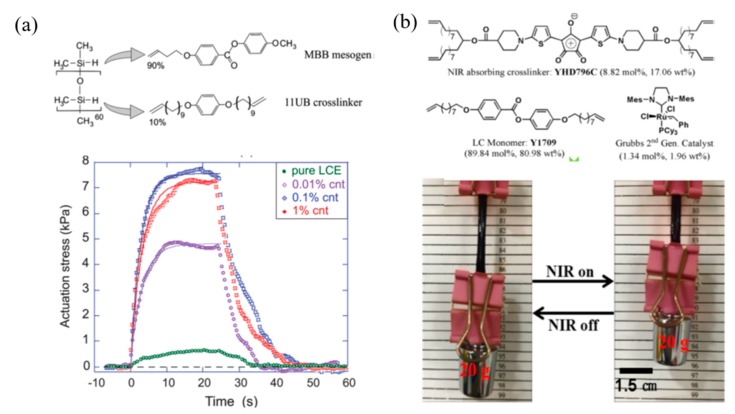
Photo responsive LCPs based on photo-heat effect. (**a**) An actuation stress measured by Iso-strain geometry on carbon nanotubes (CNTs)/LCPs composites irradiation with infrared light [71]. (**b**) The chemical structures of near-infrared spectroscopy (NIR) chromophore and LC monomers. A LCP film lifts up a load under NIR illumination irradiation. Reprinted with permission from [67]. Copyright (2017) American Chemical Society.

**Figure 6 molecules-25-01241-f006:**
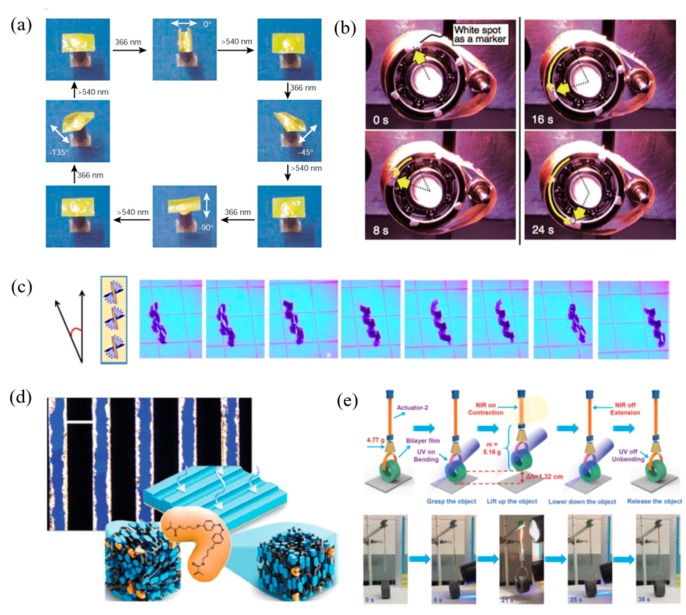
Photo responsive LCPs based on *trans–cis* photoisomerization of azobenzene irradiation by UV and visible light. (**a**) Precise control of the film bending and unbending in different directions in response to linearly polarized light. Reproduced with permission from [72]. Copyright 2003, Nature Publishing Group. (**b**) The rotation of the LCP laminated plastic motor driven by simultaneous irradiation with UV and visible light at room temperature. Reproduced with permission from [14]. Copyright © 2008 WILEY-VCH. (**c**) Photo motility of an azo-LCP strip in the twisted nematic geometry forms a spiral ribbon and continuously moves to the right. Reproduced under the terms of the CC BY 4.0 license from [75]. Copyright 2016, The Authors, published by Springer Nature. (**d**) The reversible surface change of chiral nematic surface coatings. Reproduced with permission from [11]. Copyright © 2012 WILEY-VCH. (**e**) Preciously controllable motions of a light driven robot to grasp, lift up, lower down, and release a tube. Reproduced with permission from [45]. Copyright © 2018 WILEY-VCH.

**Figure 7 molecules-25-01241-f007:**
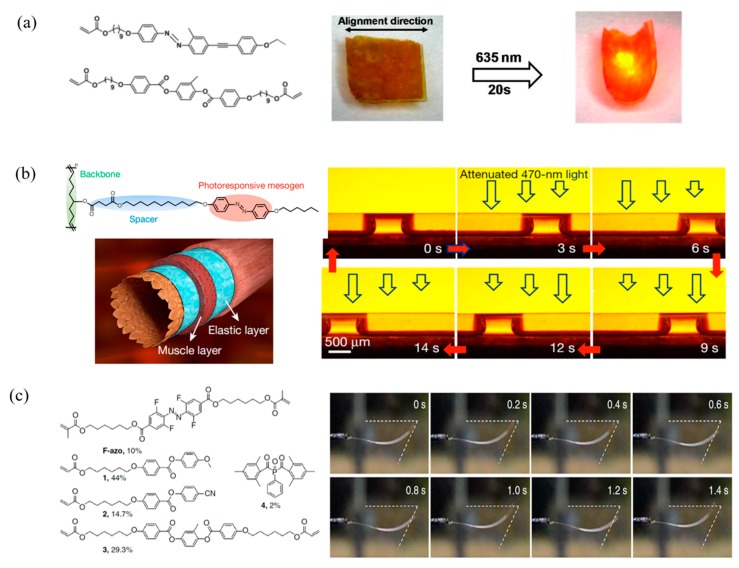
Photo responsive LCPs based on *trans–cis* photoisomerization of azobenzene irradiation by NIR light. (**a**) Chemical structures of LC monomers and the photographs of the as-prepared assembly film bending toward the 635 nm laser. Reprinted with permission from [82]. Copyright (2006) American Chemical Society. (**b**) Schematic illustration of the structure of artery walls of tubular micro-actuator, (left)). Manipulation of fluid propulsion by photo-induced tubular micro-actuators, (right). Reprinted with permission from [83]. Copyright 2016 Nature Publishing Group. (**c**) Chemical structures of ortho-fluoroazobenzene mesogens and schematic of the LC splay aligned azobenzene-containing film (left). Series of snapshots depict self-oscillatory motion of the film during sun exposure (right). Reproduced under the terms of the CC BY 4.0 license [78]. Copyright 2016, The Authors, published by Springer Nature.

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
