# Peer review of "A Review on Liquid Crystal Polymers in Free-Standing Reversible Shape Memory Materials"

_molecules, 2020, doi:10.3390/molecules25051241_

Round 1
Reviewer 1 Report
This review manuscript is well summarizing shape memory material using liquid crystalline polymers with macroscopic orientation. I recommend this manuscript will be published on Molecules.
Author Response
Dear Reviewer,
Thanks for your positive recommendation.
Reviewer 2 Report
The review "A review on liquid crystal polymers in free-standing reversible shape memory materials" authored by Zhibin Wen, Keke Yang, and Jean-Marie Raquez gives a brief summary on the current state of using liquid crystal polymers for shape memory and shape change materials and was submitted for MDPI Molecules. Some conceptional and/or structural changes are recommended before it is publishable.
Most of all for "Molecules" the whole article lacks of discussion of the effects on the molecular level. Please emphasize the difference in the presented research on the molecular level and also present more schemes with "chemistry".
When presenting chemical schemes (like Fig 2) please choose a uniform style and maximize readability.
Another important aspect is the comparison of the performance of different materials - please define all terms, which are used. For example in the manuscript different terms are used for alterations in dimensions: "change of length", "strain", "shrinkage", "contraction", etc., quantified in %, yet it is not clear whether these are ratios of measured variables or also include differences (e.g. the standard definition of strain). "shrinkage" could be volumetric or just in one dimension. At best all materials are described by the same terms (this also applies for the concept "stress").
The distiction between "thermal" and "photo" responsive LCPs does not work for me when "photothermal" effects are included in the photoresponsive section. I agree that this might work from the application point of view but on the molecular level (which should be the scope) this discussion is not suitable.
Besides, I highly recommend some additional proof reading to detect typos.
Author Response
Response to Reviewer 2 Comments
Thanks for your positive recommendation and valuable comments. The manuscript has been revised accordingly as following: Please see the revised manuscript in attachment.
Point 1: Most of all for "Molecules" the whole article lacks of discussion of the effects on the molecular level. Please emphasize the difference in the presented research on the molecular level and also present more schemes with "chemistry".
Response 1: Firstly, chemical structures of common liquid crystal monomers are summarized in Figure 1c. Secondly, more schemes with chemistry of the presented research have been added in Figure 3,4,5,7. Thirdly, the effect and the chemistry synthesis on the molecular level are discussed in the revised manuscript.
Point 2: When presenting chemical schemes (like Fig 2) please choose a uniform style and maximize readability.
Response 2: The chemical structures have been redrawn in Fig 2 in revised manuscript.
Point 3: Another important aspect is the comparison of the performance of different materials - please define all terms, which are used. For example in the manuscript different terms are used for alterations in dimensions: "change of length", "strain", "shrinkage", "contraction", etc., quantified in %, yet it is not clear whether these are ratios of measured variables or also include differences (e.g. the standard definition of strain). "shrinkage" could be volumetric or just in one dimension. At best all materials are described by the same terms (this also applies for the concept "stress").
Response 3: We have checked and defined the ratio of shape change which is compared to the original monodomain samples in whole manuscript.
Point 4: The distiction between "thermal" and "photo" responsive LCPs does not work for me when "photothermal" effects are included in the photoresponsive section. I agree that this might work from the application point of view but on the molecular level (which should be the scope) this discussion is not suitable.
Response 4: Based on the molecular level, direct heating and photothermal contribute to thermal responsive LCPs.
Point 5: Besides, I highly recommend some additional proof reading to detect typos.
Response 5: We have checked the manuscript carefully and try our best to correct the grammatical errors and typos.

Reviewer 3 Report
This manuscript comprehensively reviews regarding liquid-crystalline polymers and highlights their unique behavior on the thermal- and photo-responsive reversible shape memory effects. Overall, it is well structured and it can be interesting for broad readers working in the field of liquid crystal as well as material sciences. This reviewer would recommend to accept to publication in Molecules after minor modification mentioned below:
(1) Page 2, Line 59: Please add hyphen or blank between “polysiloxane” and “based”.
(2) Page 4, Line 125 “A similar micrometer-size actuator was shown in Figure 3f synthesized by …”: The figure (f) is lack in the Figure 3. Please provide figure (f) and suitable caption in Figure 3f.
Author Response
Responses to Reviewer 3 Comments
Thanks for your positive recommendation and valuable Points. The manuscript has been revised. Please see the attachment.
Point 1: Page 2, Line 59: Please add hyphen or blank between “polysiloxane” and “based”.
Response 1: A hyphen has been added.
Point 2: Page 4, Line 125 “A similar micrometer-size actuator was shown in Figure 3f synthesized by …”: The figure (f) is lack in the Figure 3. Please provide figure (f) and suitable caption in Figure 3f.
Response 2: This is a typo. The manuscript has been revised as Figure 3e.

Round 2
Reviewer 2 Report
Thanks to the authors for the revision. After a final spell check I can recommend to publish the article!